# Precise Authenticity of Quinoa, Coix Seed, Wild Rice and Chickpea Components Using Optimized TaqMan Real-Time PCR

**DOI:** 10.3390/foods12040852

**Published:** 2023-02-16

**Authors:** Qiuyue Zheng, Xinying Yin, Aifu Yang, Ning Yu, Ranran Xing, Ying Chen, Ruijie Deng, Jijuan Cao

**Affiliations:** 1Key Laboratory of Biotechnology and Bioresources Utilization of Ministry of Education, College of Life Science, Dalian Minzu University, Dalian 116600, China; 2Technology Center of Dalian Customs District, Dalian 116001, China; 3Chinese Academy of Inspection and Quarantine, Beijing 322001, China; 4Healthy Food Evaluation Research Center, College of Biomass Science and Engineering, Sichuan University, Chengdu 610065, China

**Keywords:** food authenticity, quinoa, coix seed, wild rice, TaqMan real-time quantitative polymerase chain reaction

## Abstract

Functional food such as, quinoa, coix seed, wild rice and chickpea have experienced rapidly increasing demand globally and exhibit high economic values. Nevertheless, a method for rapid yet accurate detection of these source components is absent, making it difficult to identify commercially available food with labels indicating the presence of relevant components. In this study, we constructed a real-time quantitative polymerase chain reaction (qPCR) method for rapid detection of quinoa, coix seed, wild rice and chickpea in food to identify the authenticity of such food. Specific primers and probes were designed with 2S albumin genes of quinoa, SAD genes of coix seed, ITS genes of wild rice and CIA-2 genes of chickpea as the target genes. The qPCR method could specifically identify the four wild rice strains, yielding, LODs of 0.96, 1.14, 1.04 and 0.97 pg/µL quinoa, coix seed, wild rice and chickpea source components, respectively. Particularly, the method allowed the identification of the target component with content below 0.01%. A total of 24 commercially available food samples of different types were detected by using the method and the results indicate that the developed method is applicable to the detection of different food matrices, as well as authenticity verification in deeply processed food.

## 1. Introduction

Modern food nutrition is developing rapidly; functional food with high nutrition and high value meets the current public demand for nutritious and healthy food, has excellent nutritional quality and high commercial value, and its consumption is growing rapidly [1,2]. Quinoa (*Chenopodium quinoa Willd*.), coix seed (*Coix lacryma-jobi*), wild rice (*Zizania latifolia*, *Zizania palustris*, *Zizania aquatica*, *Zizania texana*) and chickpea (*Cicer arietinum* L.) are emerging as functional foods, growing, producing and processing increasingly commercially, and have become a high nutritional value, unique flavor, expensive healthy food in the market. Quinoa comprises three strains, white quinoa, black quinoa and red quinoa [3]. Quinoa is native to the Andean region and is becoming an increasingly important food crop due to its unique nutritional and health values [4]. The Food and Agriculture Organization of the United Nations (FAO) has recognized quinoa as a unique plant that can meet all human nutritional needs by itself and has promoted and publicized quinoa [5,6]. Coix seed is an important small grain in Asian countries and an important cash crop widely grown in Southeast Asia [7,8], containing protein, fat, multiple vitamins and trace elements, with high nutritional and medicinal value [9]. Wild rice, is rich in protein, essential amino acids, dietary fiber and various trace elements with low-fat content [10,11,12]. Wild rice has become a high-value health food, known as the caviar of grains [13]. Chickpea is one of the most consumed legumes in the world, grown and consumed worldwide, with more than 2.3 million tons entering the world market every year [14,15]. Chickpea has become a popular newly developing emerging plant-based food. 

Owing to their high nutritional and economic values, quinoa, coix seed, wild rice and chickpea are easy targets of food adulteration in international trade and market circulation, and the phenomenon of food adulteration has become a global problem for illegal traders to make profits [6]. Since the bulk of the population has no experience in the morphological identification of plant species, especially when processed into powder or food, it is impossible to identify them morphologically. Nevertheless, a method for rapid yet accurate detection of quinoa, coix seed, wild rice and chickpea is absent, making it difficult to identify commercially available food with labels indicating the presence of relevant components. Hence, a rapid yet sensitive authenticity identification method is urgently needed. Additionally, raw materials components of processed food are complex, and most processed foods are heated at high temperatures, resulting in protein denaturation and biological enzyme inactivation, which cannot be used as indicators of food authenticity. In view of the complexitiy of food fraud and its strong avoidance of detection, it is necessary to establish a practical and effective method for food authenticity identification [16].

Current identification methods for food authenticity include morphological analysis and sensory analysis, proteomics, metabolomics based on physical and chemical analysis, DNA-based genomics and sensor nondestructive testing [17]. Earlier studies mostly used morphological identification, microscopic identification and sensory identification, with the problem of unreliable identification results, which can no longer meet the current food adulteration safety control need. Proteomics techniques include protein chips, ELISA, isotope labeling, electrophoresis, liquid chromatography and mass spectrometry [18,19]. Metabolomics is an emerging research tool based on the analysis of all small molecule metabolites in the bio-system for non-specific target substance screening [20,21,22]. The proteomics and metabolomics assays require expensive and large instrumentation and the construction of comprehensive and accurate databases, limiting the wide application of these technologies.

With the continuous development of molecular biology technology—nucleic acids are more stable than proteins and are rich in genetic information—DNA-based molecular biology methods are widely used in food detection. Among them, real-time quantitative polymerase chain reaction (qPCR) has become an important tool for food authenticity detection because of its high specificity, sensitivity, reproducibility, speed and fully closed reaction [23,24]. Wu et al. developed a qPCR-based method for the identification of honey species and adulteration detection for six honey species produced in China, and the limit-of-detection (LOD) of adulterated honey could reach 0.1–0.5% [25]. Garrido-Maestu et al. evaluated the ability of qPCR and real-time loop-mediated isothermal amplification (qLAMP) to detect and quantify gluten, and qPCR was more sensitive than qLAMP [26]. In order to evaluate the effect of DNA fragmentation in deeply processed food on the reliability of qPCR analysis, Mano et al. investigated a new method to quantify the degree of DNA fragmentation by creating a DNA fragmentation index (DFI) as an index value, and demonstrated the validity and reliability of the qPCR method in the quality control of DNA detection in deeply processed food by evaluating the relationship between DFI and LOD [27]. In summary, qPCR is suitable for the identification of complex matrix and deeply processed food species in honey, herbal medicine, feed, etc. The method is specific and sensitive, and widely used; however, the authenticity identification of novel functional food of plant origin is a new hot issue that needs attention.

In view of the obvious advantages of the qPCR method in food authenticity and adulteration detection, and considering the fact that the deep processing process of food may lead to the breakage of genome DNA into small fragments, which affects the efficiency of PCR detection, this study selected conserved gene segment as short and specific as possible as the target gene for species identification. The specific primer and probe were designed and the reaction parameters were optimized to develop a qPCR method for rapid detection and authenticity identification of quinoa, coix seed, wild rice and chickpea in deeply processed food. It provides technical support for ensuring food quality and safety, identifying and distinguishing economically motivated adulteration, protecting consumers from economic fraud, and maintaining a fair market environment. 

## 2. Material and Methods

### 2.1. Materials

Quinoa, coix seed, wild rice and chickpea samples were purchased from a local seed dealer and food containing such components were purchased from local and online supermarkets. Other plant seed samples for specificity detection, including corn, sorghum rice, pseudo sorghum rice, chia seed, red rice, black rice, rice, yellow rice, rye, buckwheat, tartaric buckwheat, oats, barley, wheat, black sesame, black beans, kidney beans, soybeans, mung beans, lentils, peas, red beans, etc., were purchased from seed dealers in Dalian, Liaoning Province, China. Edible fungi and shiitake mushrooms were purchased from a local supermarket in Dalian, China. The ITS gene of the purchased target and other plant seed samples were sequenced to confirm the species of seed samples (TaKaRa Co., Ltd., Dalian, China).

### 2.2. Extraction of Genome DNA

The seeds of planting samples were ground in liquid nitrogen. Food samples were homogenized using a high-speed tissue grinder (8010G, Waring, Stamford, CT, USA). A total of 100 mg of the crushed sample was taken and 1.5 mL of CTAB buffer (cetyltrimethylammonium bromide, CTAB) containing 55 mM CTAB, 1400 mM NaCl, 20 mM EDTA (ethylene diamine tetra acetic acid), 100 mM Tris was added; 10 µL of proteinase K (20 mg/µL) was added. After shaking and mixing with a vortex oscillator, it was incubated at 65 °C for 30 min. Then, the mixture was centrifuged at 12,000 rpm for 5 min, the supernatant was removed, and 400 µL trichloromethane: isoamyl alcohol (24:1) was added, uniformly mixed and centrifuged at 12,000 rpm for 5 min, the supernatant was removed, 0.8 times the volume of isopropanol was added, and it was precipitated at room temperature for 1 h–2 h. Then it was centrifuged at 12,000 rpm for 10 min, the supernatant was discarded, it was washed once with 70% ethanol, and air dried. The DNA was dissolved by adding 50 µL TE (10 mM Tris-HCl, pH 8.0, 1 mM EDTA, pH 8.0), and stored at −20 °C.

### 2.3. qPCR Primer and Probe Design

The quinoa 2S albumin gene, coix seed stearoyl-acyl-carrier protein desaturase gene (SAD) gene, wild rice ribosomal internal transcribed space (ITS) gene and chickpea chloroplast import apparatus-2 gene (CIA-2) gene sequences were retrieved from the National Center for Biotechnology Information (NCBI) database of USA as search templates for BLAST comparison. In addition, DNA sequences of other species were retrieved for the specificity test. DNA sequences were compared using MEGA 4.0 to select DNA fragments with high variability as the target gene for amplification. Primer 5.0 was used to design specific primers and probes. The eukaryotic 18SrRNA universal primer and probe were used as internal reference genes. The eukaryotic 18SrRNA universal system was used separately as a single system. The primer and probes were synthesized by TaKaRa Co., Ltd. (Dalian, China). 

### 2.4. Real-Time qPCR 

The qPCR analysis was performed on a QuantStudio6 Flex real-time quantitative fluorescence PCR instrument (ABI, Waltham, MA, USA). The reaction system was 25 µL, including 12.5 μL of the qPCR reaction mix (2×) (Premix Ex TaqTM Probe qPCR, RR390 A, TaKaRa, Dalian, China), 1 μL of forward and reverse primers (10 μM), 1 μL of probe (5 μM), 2 μL of DNA template (10–100 ng/μL), and 8.5 μL of sterilized water. The qPCR thermal cycling program was 95 °C pre-denaturation for 10 min, followed by 40 cycles of 95 °C for 5 s and 60 °C for 30 s.

### 2.5. Specificity Test

Genome DNA for specific detection was extracted from planting seed samples. The amplification of the synthetic universal eukaryotic 18SrRNA primer was used to verify whether it was suitable for qPCR detection. Then, qPCR reactions were carried out using specific primers and probes of quinoa, coix seed, wild rice and chickpea respectively, and the specificity and cross-reactivity of the qPCR method was analyzed.

### 2.6. Sensitivity and Amplification Efficiency Test

The limit of detection (LOD) is an important parameter for evaluating the effective detection range and accurate quantification range of a method, and is the lowest concentration of target samples that can be detected by the analytical process at the 95% confidence interval (CI) [28,29]. Wheat is a common food matrix, and wheat genome DNA (10 ng/μL) was used as non-target background DNA, as a gradient dilution of genome DNA of quinoa, coix seed, wild rice and chickpea; DNA solutions containing 1.0 ng/μL, 0.1 ng/μL, 0.01 ng/μL, 1.0 pg/μL, 0.5 pg/μL, 0.2 pg/μL and 0.1 pg/μL target samples were prepared to determine LOD, and 12 sub-samples of each dilution gradient were tested in parallel. Probabilistic regression analysis was performed on the data using MedCalc Software (MedCalc Software bvba, Ostend, Belgium) to calculate the LOD95% of the qPCR assay.

Genome DNA extracted from four target samples (adjusting concentration to 100 ng/μL with TE) was serially diluted 10-fold to 10 ng/μL, 1.0 ng/μL, 0.1 ng/μL, 0.01 ng/μL, 1.0 pg/μL, and 0.1 pg/μL, respectively. The DNA of six concentrations was used as the template, and each concentration was repeated three times to investigate the efficiency of the qPCR reaction. The linear regression analysis was performed by plotting a linear standard curve with the logarithm of the concentration of template DNA as the horizontal coordinate and the average Ct value of the amplification of each concentration as the vertical coordinate. The linear correlation coefficient (R^2^) values were calculated from the equation of the standard curve and its slope. The qPCR efficiency (E) was calculated using E = 100 (10–1/slope-1) and expressed as a percentage [28].

### 2.7. Robustness Evaluation

Reaction equipment and reaction parameters used for qPCR reactions will vary from laboratory to laboratory. In order to assess whether changes in experimental conditions affect the performance of the method, the orthogonal design was used to analyze the robustness of the qPCR method by varying the qPCR equipment, reagents, primer and probe concentrations, reaction mixture volume and annealing temperature according to the qPCR validation guidelines [28]. Among them, qPCR instruments, using LC 480 II (Roche Diagnostics, Swiss Confederation, Basel, Switzerland) and ABI Q6 (ABI, Los Angeles, CA, USA); qPCR reagents, using Premix Ex Taq^TM^ Probe qPCR (RR390 A, TAKARA, Beijing, China) and GoTaq qPCR Master Mix (Cat. A6101, Promega, Fitchburg, WI, USA); primer and probe concentrations (−20%), reaction mix volume (2×) (±1 µL), annealing temperature (±1 °C). The DNA concentration of 3 × LOD was used as a template and each analysis was repeated three times. The parameter variations were evaluated using the orthogonal design (Appendix A, in the Appendix A). The standard deviation (SD) and the repeatability standard deviation (RSD) of the robustness test results were calculated for different combinations of orthogonal designs [28,29].

### 2.8. Detection of Commercial Samples

A total of 24 commercial products containing quinoa, coix seed, wild rice and chickpea source in the detection label were collected, covering common food categories, including seed, semi-processed products (e.g., broken rice, powder, flour), deeply processed food (e.g., porridge, steamed bread, biscuit, cereal, solid tea and canned food). There were 10 species products containing coix seed, 7 species containing chickpea, 5 species containing quinoa, and 2 species containing wild rice. Commercial product testing was used to further evaluate the effectiveness of the qPCR method and to determine whether food fraud was involved in the commercial practices by comparing the goods labels with the qPCR test results. 

### 2.9. Quantification Performance of qPCR Methods

To assess the detection performance of qPCR methods established in this paper, a series of different proportions of seed powder mixtures (0–10%) of target samples to wheat was prepared. Genome DNA of mixture powder samples with different concentrations was extracted to determine the detection quality of the qPCR system established in this study in the mixed matrix.

## 3. Result and Discussion

### 3.1. Specific Analysis of Primer and Probe

After gene sequences comparison, the highly conserved 2S albumin gene [30,31] (GenBank: XM_021902904.1) of quinoa, SAD gene [32,33] (GenBank: MK589804.1) of coix seed, ITS genes [34] of four species of wild rice (GenBank: AF169234.1, AF169232.1, AF169231.1, AF169233.1) and CIA-2 gene [35] of chickpea (GenBank: XM_004507840.3) were selected as the target genes, respectively (Figure 1 and Appendix A). Sequences of the proposed primers and probe were listed in Table 1. The FAM fluorescent reporter motif was labeled at the 5′ end of the probe, and the BHQ1 fluorescent quenching motif was labeled at the 3′ end.

### 3.2. Specificity Tests

The eukaryotic 18SrRNA universal primer was used to amplify the extracted plant genome DNA, and all samples showed amplification curves (Ct value 13.6–18.8), indicating that there were no pollutants in the extracted DNA samples that inhibited PCR reaction, and they were all suitable for qPCR detection. Genome DNA of the 24 products containing quinoa, coix seed, wild rice or chickpea, and other non-target plants were tested in three replicates using a specific primer and probe. Only four wild rice strains, three quinoa strains (white, red and black), barley and chickpea samples showed positive results (Appendix A, in the Appendix A), with no amplification of genome DNA of other non-target plants such as rice, and no cross-reactivity between different species, proving the specificity of qPCR method.

### 3.3. Sensitivity Test

Quinoa, coix seed, wild rice and chickpea genome DNA were analyzed after serial dilutions using wheat seeds genomic DNA. A total of 12 replicates of each concentration gradient were analyzed, and positives were detected in 12 parallels at a 0.001 ng/µL (1000-fold dilution gradient) concentration level. Probabilistic regression analysis was performed with MedCalc Software to obtain the limit of detection (LOD) of qPCR under 95% confidence intervals (CI). The LOD of the quinoa source component was 0.96 pg/μL with 95% CI of 0.69–2.12 pg/μL, the coix seed source component was 1.14 pg/μL with 95% CI of 0.68–3.85 pg/μL, the wild rice source component was 1.04 pg/μL with 95% CI of 0.71–2.42 pg/μL and the chickpea source component was 0.97 pg/μL with 95% CI of 0.65–2.30 pg/μL under 95% CI. The results demonstrate that quinoa, coix seed, wild rice and chickpea source components could be detected by the developed qPCR method in products containing only small amounts of seed below 0.01% (Figure 2).

A linear standard curve was plotted for the qPCR amplification results of the six dilution concentrations of target genome DNA, and a linear relationship between Ct value and the logarithm of DNA concentration was detected (Figure 2). The calculated linear equation is shown in Figure 2, and the correlation coefficients (R^2^) of the quinoa, coix seed, wild rice and chickpea qPCR assays were 0.9923, 0.9984, 0.9965 and 0.9989, respectively (Appendix A, in the Appendix A). The amplification efficiency values (E) of quinoa, coix seed, wild rice and chickpea were 102.22%, 98.74%, 96.97% and 97.80%, which meets the requirements of “R^2^ ≥ 0.99 and E of 90–110%” in the general qPCR validation guidelines [28], confirming that the developed qPCR method shows high amplification efficiency.

### 3.4. Quantification Performance of qPCR Methods

Genome DNA was extracted from the mixture containing 10%, 1%, 0.1%, 0.01% and 0.001% of the target component (including quinoa, coix seed, wild rice or chickpea) for qPCR amplification. The mixtures containing 10%, 1%, 0.1% and 0.01% of the target component were detected; 0.001% were undetected, like the negative control (Figure 3). Therefore, the qPCR reaction system can detect the components from quinoa, coix seed, wild rice or chickpea as low as 0.01% (*w*/*w*) of the mixture. These results prove that the developed qPCR methods are applicable to mixture types of food, including the detection of the target component as low as 0.01%.

### 3.5. Robustness Test

The concentration of template DNA in the reaction system remained unchanged, and the parameters of qPCR such as equipment, reagents, primer and probe concentrations, the volume of the reaction mixture and annealing temperature were changed for conducting the qPCR, respectively. The orthogonal test results of each combination were statistically analyzed. The Ct value of quinoa was 35.4 ± 0.05 and the reproducibility standard deviation (RSD) was 0.14%, coix seed was 35.93 ± 0.06 and the RSD was 0.17%, wild rice was 32.86 ± 0.16 and the RSD was 0.49%, chickpea was 35.91 ± 0.21, and the RSD was 0.58% (Figure 4 and Appendix A). According to these results of the robustness test, the RSD of the qPCR systems for quinoa, coix seed, wild rice and chickpea was less than 1%, which is far below the requirement of “RSD should not exceed 30% for each combination of changes” in the qPCR validation guidelines [29]. From these results, it can be concluded that the qPCR detection systems are stable and reliable, and developed qPCR methods are suitable for different laboratories and testing conditions.

### 3.6. Analysis of Commercial Samples

A total of 24 representative commercial food samples with labels indicating the presence of quinoa, coix seed, wild rice and chickpea, including baking food, canned food, and instant food, were tested in this study. All samples were able to detect the 18SrRNA internal reference gene, which suggested that DNA extracted from different food matrices is applicable for qPCR amplification. Herein, the 10 types of food containing coix seed were detected, including blended powder, biscuit, pellet, cream, solid tea, fudge, noodles, steamed buns, etc. Coix seed was detected in sample numbers 1, 2 and 5 to 10. There were two food samples (No. 3 and 4) containing coix seed in the tables but the detection result was negative. Chickpea was detected in canned food and fried food (No. 11 to 17) which were labeled as containing chickpeas. All seven species' goods testing results were positive. Quinoa goods (No. 18 to 22) were analyzed by the qPCR method. The result showed that the sample named “Quinoa soda biscuit (3.2%)” (No. 20) never detected the component of quinoa and the other four types of goods detection results were positive. Wild rice products (No. 23 and 24) were tested. The results of both two samples contained wild rice. These results confirm that most commercial samples contain the target component of the labels, whereas a few samples were determined uncertainly, and were suspected of food fraud (Figure 5 and Appendix A). 

Products were baked, heated, and autoclaved, which may cause degradation of sample genome DNA, break into fragments of different lengths, reduce the sensitivity of qPCR, or contain large amounts of fat and protein that interfere with the qPCR reaction. Considering the extreme condition that has the greatest impact on DNA during the processing of canned food or thermally processed food is thermal sterilization, we focused on the detection of high-temperature processed goods and detected the identified target component. These results prove that the developed qPCR method is applicable to various types of food, including the detection of the target component in deeply processed food.

## 4. Conclusions

In summary, a novel functional food qPCR species identification method to check for commercial fraud was developed. To assess the authenticity of food containing quinoa, coix seed, wild rice and chickpea, we developed a qPCR reaction system to detect the seeds of quinoa, coix seed, wild rice and chickpea in food. The LOD of quinoa was 0.96 pg, coix seed was 1.14 pg, wild rice was 1.04 pg and chickpea was 0.97 pg, which allowed the detection of products containing less than 0.01% of the seed. A total of 24 commercial food samples with labels indicating the presence of quinoa, coix seed, wild rice and chickpea, including baking food, canned food, and instant food were analyzed by qPCR, and corresponding target components were detected. A total of 21 samples were consistent with the commodity labels; two coix seed red bean meal substitutes and red bean coix seed pellets with coix seed identification did not contain a coix seed component and one quinoa soda biscuit sample with the identification of quinoa did not contain quinoa component. This can be attributed to the sale of counterfeit products in e-commerce retail, suspected of food fraud. QPCR methods developed for the detection of quinoa, coix seed, wild rice and chickpea materials described here are specific, sensitive, robust, and facilitate the production and consumption of novel functional food and protect food safety and the interest of consumers.

## Figures and Tables

**Figure 1 foods-12-00852-f001:**
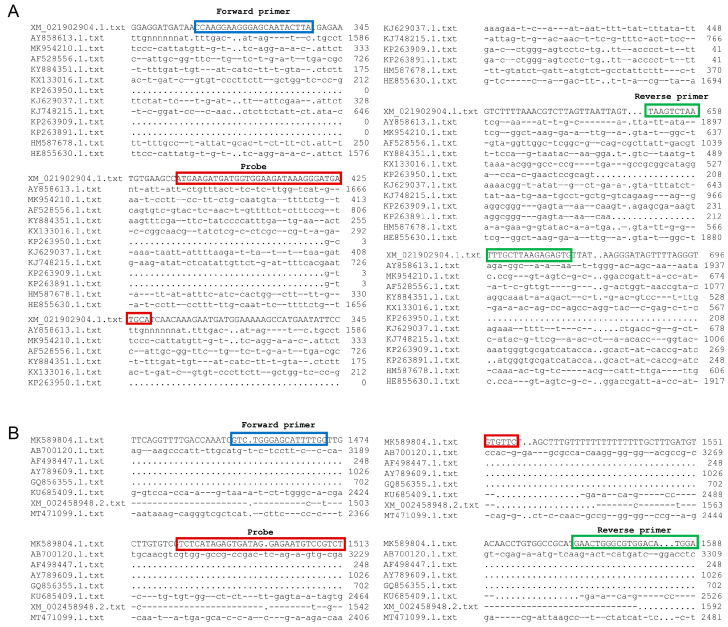
Specific analysis of primer and probe. (**A**) Specificity comparison of primer and probe sequences of the 2S albumin gene region developed for specific qPCR detection of quinoa source component using MEGA 4.0. (**B**) Specificity comparison of primer and probe sequence of SAD gene region developed by specific qPCR detection of coix seed source component using MEGA 4.0.

**Figure 2 foods-12-00852-f002:**
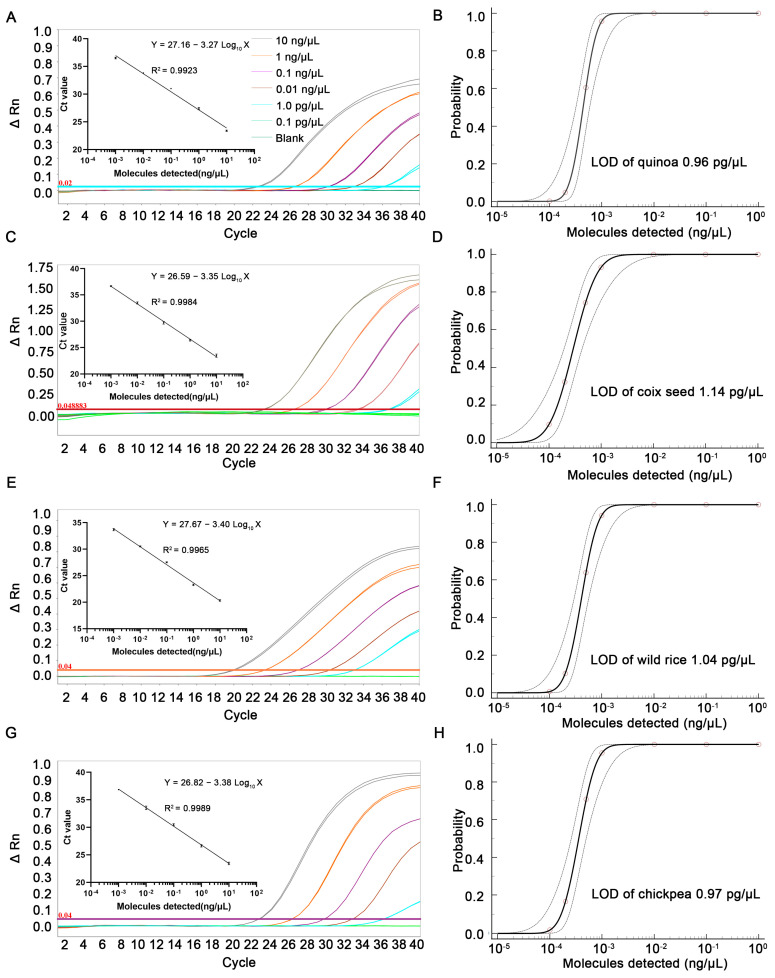
Amplification curves and standard curves for the analysis of 6 dilution steps of quinoa (**A**), coix seed (**C**), wild rice (**E**), chickpea (**G**) qPCR assay. Inner: standard curves and linear equation of quinoa (**A**), coix seed (**C**), wild rice (**E**) and chickpea (**G**), respectively. Probability regression curves of LOD of quinoa (**B**), coix seed (**D**), wild rice (**F**) and chickpea (**H**) by qPCR assay. Probabilistic regression analysis was performed with MedCalc software to obtain the LOD of qPCR under 95% CI. The LOD of quinoa source component was 0.96 pg/μL (**B**); the LOD of coix seed source component was 1.14 pg/μL (**D**); the LOD of wild rice source component was 1.04 pg/μL (**F**); the LOD of the chickpea source component was 0.97 pg/μL (**H**).

**Figure 3 foods-12-00852-f003:**
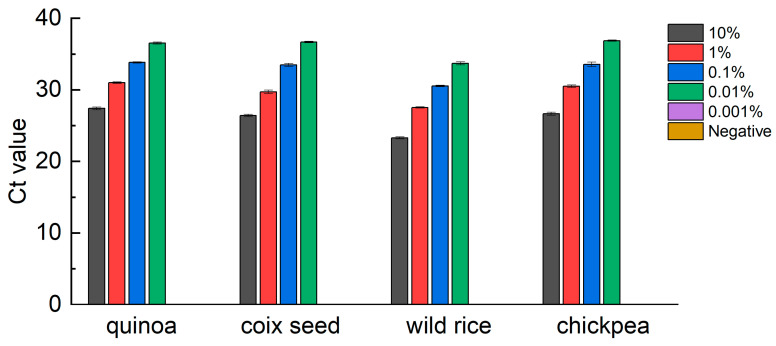
Quantification performance of qPCR methods. The Ct value of qPCR method corresponding to the addition of different percentages of quinoa, coix seed, wild rice or chickpea added into mixture of wheat ranging from 0% to 10% (0%, 0.001%, 0.01%, 0.1%, 1% and 10%). 0.001% and negative control were undetected.

**Figure 4 foods-12-00852-f004:**
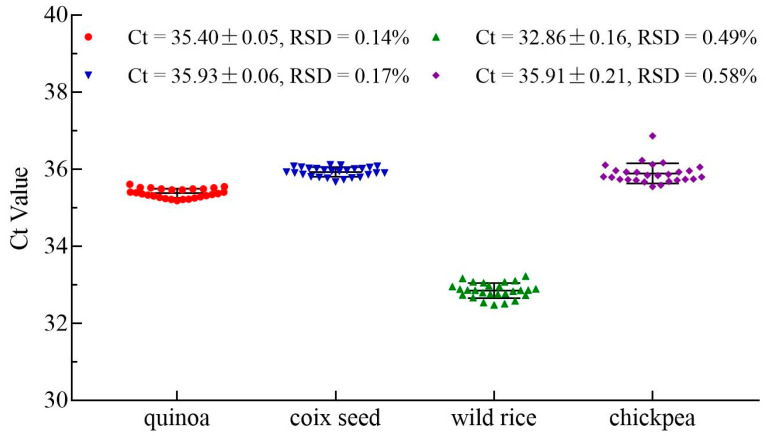
Robustness of quinoa, coix seed, wild rice and chickpea systems. The Ct value ± SD of quinoa was 35.4 ± 0.05, and the RSD was 0.14%; the Ct value of coix seed was 35.93 ± 0.06, and the RSD was 0.17%; the Ct value of wild rice was 32.86 ± 0.16, and the RSD was 0.49%; the Ct value of chickpea was 35.91 ± 0.21, and the RSD was 0.58%.

**Figure 5 foods-12-00852-f005:**
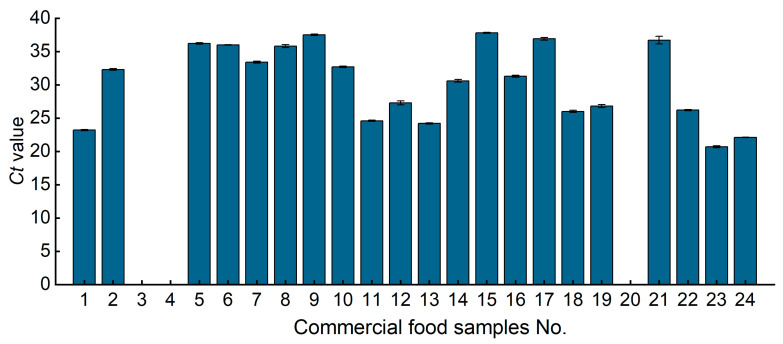
Detection of commercial food samples containing quinoa, coix seed, wild rice and chickpea. The number of samples were consistent with the “Food samples’ name” in Appendix A. A total of 24 commercial food samples with labels indicating presence of coix seed (No. 1–10), chickpea (No. 11–17), quinoa (No. 18–22) and wild rice (No. 23 and 24), including baking food, canned food, and instant food were analyzed by qPCR, and corresponding target components were detected. A total of 21 samples were consistent with the commodity labels; two coix seed samples (No. 3 and 4) were identified that did not contain coix seed component and one quinoa sample (No. 20) did not detect a quinoa component.

**Table 1 foods-12-00852-t001:** Primers and probes sequence used in this study ^a^.

Species	Target Gene	Primer and Probe	Sequence (5′ → 3′)	Position	Size (bp)
quinoa	2S albumin	forward primer	CCAAGGAAGGGAGCAATACTTAG	319–341	357
		reverse primer	ACACTCTCTTAAGCAAATTAGACTTAAC	648–675	
		probe	FAM-ATGAAGATGATGGTGGAAGATAAAGGGATGATGCA-BHQ1	395–429	
coix seed	SAD	forward primer	GTCTGGGAGCATTTTGCTTGC	1455–1475	134
		reverse primer	TCCATGTCCACGCCCAGTTC	1569–1588	
		probe	FAM-TCTCATAGAGTGATAGGAGAATGTCCGTCTGTGTTC-BHQ1	1484–1519	
wild rice	ITS	forward primer	CGAGAGTCGTGTGGATGTTGT	227–247	238
		reverse primer	TGCGGAAGGATCATTGTCGT	10–29	
		probe	FAM-CGGCGGTCGGTAAGAGGTGTTCC-BHQ1	175–197	
chickpea	CIA-2	forward primer	AGAAGAAGGTTGTTACGGTGGAG	1422–1444	170
		reverse primer	CGGTGCGTCGGAGATAGGA	1572–1591	
		probe	FAM-GAAGGCGTTCGGAATGCTTGGTCT GATAAA-BHQ1	1520–1548	
Reference	18SrRNA	forward primer	TCTGCCCTATCAACTTTCGATGGTA	233–257	137
		reverse primer	AATTTGCGCGCCT GCTGCCTTCCTT	345–369	
		probe	FAM-CCGTTTCTCAGGCTCCCTCTCCGGAATCGAACC-BHQ1	290–322	

^a^ FAM: 6-carboxy-fluorescein, BHQ1: Black Hole Quencher1.

## Data Availability

Data is contained within the article and Appendix A.

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
