# Peer review of "Precise Authenticity of Quinoa, Coix Seed, Wild Rice and Chickpea Components Using Optimized TaqMan Real-Time PCR"

_foods, 2023, doi:10.3390/foods12040852_

Round 1

Reviewer 1 Report

The manuscript entitled “Precise and rapid authenticity of functional food components using optimized TaqMan real-time quantitative PCR” is a report to identify the quinoa, coix seed, wild rice and chickpea in commercial food using TaqMan real-time PCR assays.  I think the topic of the manuscript is interesting and important. Overall, the manuscript is well-written and very thorough. Therefore, I recommend the following corrections:

Lines 2-3: The authors developed qPCR assays for the detection of each target species separately (as singleplex) but not for simultaneous (multiplex) detection. Therefore, it would not be reasonable to mention it as a rapid method. Consequently, the word ‘rapid’ should be omitted from the title.

Line 43: “andpublicizedquinoa” -is it one word?

Lines 52-54: “Owing to their high nutritional and economic values, quinoa, ………………………………………… become a global problem for illegal traders to make profits.” -authors should include the references or evidence of this statement.

Lines 110-117: The authors purchased target and other plant seed samples from a local seed dealer. How did the authors ensure that the purchased or analysed samples were from the correct species?

Lines 146-147: 2 μL of DNA template (10-100 μg/mL)- why authors optimized the assay with μg template concentration level, however, the sensitivity and efficiency of the assays were tested with ng level template- please clarify.

Lines 150-160: “Wheat is a common food matrix, wheat genome DNA was used as non-target background DNA,”- please mention the concentration of the background DNA.

There is no ‘results and discussion’ section and also did not mention the 3.1 sec.

Line 212: Table 1. “Primers and probes sequence used in this study a”- why letter ‘a’?

Lines 246-247: “A linear standard curve was plotted for the qPCR amplification results of the six dilution concentrations of target genome DNA,”- however, Figure 2 clearly shows that the authors used 5 dilution concentrations in the standard curve - please clarify.

Line 296: Table 5 will be table S5.

The authors used the eukaryotic 18SrRNA universal system as a control, however, they did not specify whether it was used as a duplex system with each target assay or separately as a single system.

Any developed system should be validated under known samples prepared by the laboratory to ensure system reliability. Herein, the authors tested commercial food samples using the developed system. However, the system did not validate under spike food samples. Therefore, the authors should validate the system under laboratory-generated spike samples.

Reviewer 2 Report

The manuscript “Precise and rapid authenticity of functional food components using optimized TaqMan real-time quantitative PCR” aims to describe four qPCR assays for rapid detection of quinoa, coix seed, wild rice (Zizania) and chickpea to verify the food authenticity.

The topic  is of interest, but the presentation must be reviewed.

Major comments:

The grammar must be checked since some mistakes are present throughout the manuscript.

Lines 2-3: Title: In my opinion, the title is too generic. As reported in Foods | Instructions for Authors (mdpi.com) “The title of your manuscript should be concise, specific and relevant”.

Lines 109-117: the list of plant seed samples is not complete. In the text it is not reported the reference to the table S2 of supplementary materials, please add.

Lines 116-117: I don’t understand why the authors decided to test edible fungi and shiitake mushrooms, please, explain.

Line 192: 24 species? Or 24 products? Please, explain or correct.

Line 217: 25 species? Or 24 products? I don’t understand. Please, explain or correct.

Line 225: wheat? Or seeds? Wheat is Titicum aestivum or Triticum durum. Please, explain or correct.

Line 342.  As reported in Foods | Instructions for Authors (mdpi.com) Journal Articles must be written like this: Author 1, A.B.; Author 2, C.D. Title of the article. Abbreviated Journal Name YearVolume, page range.

References 11, 16, 18, 19, 28, 29, 33, 34: please, write the Journal Name in abbreviated form.

References 30 and 33: please, complete the authors name.

Table S2: Please, write Latin name of species in italics.

Round 2

Reviewer 1 Report

The revised manuscript by Zheng et al. reports on the TaqMan qPCR method to authenticate the quinoa, coix seed, wild rice and chickpea in commercial food. The revised version has added important information. However, the following modification is still needed before publication:

The authors addressed “point 12” on the response sheet only, but did not include it in the manuscript. The authors should include this experiment in the methodology and result sections.

Author Response

Point: The authors addressed “point 12” on the response sheet only, but did not include it in the manuscript. The authors should include this experiment in the methodology and result sections.

Response: Thank you for your kind suggestion. We have supplied 'point 12' experiment in methodology and result sections of the manuscript.

Reviewer 2 Report

In my opinion, the paper can be published.

Author Response

Thank you very much for your kind opinion.